# Exploring the Impact of Backward and Forward Locomotor Treadmill Training in Chronic Stroke Survivors with Severe Post-Stroke Walking Impairment: A Single-Center Pilot Randomized Controlled Trial

**DOI:** 10.3390/brainsci15050437

**Published:** 2025-04-24

**Authors:** Saiprasad Naidu, Khwahish Singh, Tamiel Murray, Colin Drury, Erin Palermo, Heidi J. Sucharew, Changchun Xie, Pierce Boyne, Kari Dunning, Oluwole O. Awosika

**Affiliations:** 1Department of Neurology and Rehabilitation Medicine, University of Cincinnati, Cincinnati, OH 45267, USA; naidusd@mail.uc.edu (S.N.); khwahish.singh2612@gmail.com (K.S.); murray.182@wright.edu (T.M.); drurycd@ucmail.uc.edu (C.D.); erincpalermo@yahoo.com (E.P.); 2Wright State University Boonshoft School of Medicine, Dayton, OH 45324, USA; 3Department of Emergency Medicine, University of Cincinnati, Cincinnati, OH 45267, USA; sucharhj@ucmail.uc.edu; 4Department of Biostatistics, Health Informatics and Data Sciences, University of Cincinnati, Cincinnati, OH 45267, USA; xiecn@ucmail.uc.edu; 5Department of Rehabilitation, Exercise and Nutrition Sciences, College of Allied Health Sciences, University of Cincinnati, Cincinnati, OH 45267, USA; boynepe@ucmail.uc.edu (P.B.); dunninkk@ucmail.uc.edu (K.D.)

**Keywords:** severe walking impairment, severe stroke, sensorineural integration, gait, outcome measures, stroke recovery, stroke rehabilitation, backward walking, dynamic postural stability, walking capacity

## Abstract

Background: Defined as a self-selected speed of <0.4 m/s, chronic stroke survivors falling in this category are classified as “severe”, usually homebound and sedentary, and they experience worse outcomes. Limited rehabilitation strategies are available to improve walking speed and related outcomes in this subgroup, and questions regarding effective rehabilitation options remain. The objective of this study was to determine the effects of backward (BLTT) and forward (FLTT) locomotor treadmill training on overground walking speed, spatiotemporal symmetry, and dynamic postural stability. Methods: In this single-center, assessor-blinded, randomized controlled pilot trial, 14 stroke survivors with severe waking impairment underwent 12 sessions of BLTT (*n* = 7) or FLTT (*n* = 7). The primary outcome was the proportion of participants reaching clinically meaningful important difference (MCID) on the 10-meter walk test following training completion. Secondary outcomes were between-group differences in walking speed, spatiotemporal symmetry, and completion time on the 3-meter timed up and go (3M TUG) at 24 h, 30 days, and 90 days POST. Results: Two subjects in the BLTT group (28.6%) and one (14.3%) in FLTT achieved MCID following training; however, most subjects did not, with significant variability in response. At 24 h POST, the median (IQR) percent change in walking speed was 28.9 (9.01–36.7) and 17.4 (12.6–39.7) with BLTT and FLTT, respectively; however, no between-group differences were seen (*p* = 0.80) at this time point or at 30 (*p* > 0.99) and 90 (*p* > 0.99) days follow up. Likewise, there were no significant between-group differences in spatiotemporal symmetry and the 3M TUG across time points. Conclusions: While preliminary, this study found that 12 training sessions did not lead to group-level achievement of MCID for walking speed in our cohort and found no significant between-group differences in walking capacity or dynamic postural stability. Future well-powered dosing trials and mechanistically driven studies are needed to optimize and identify predictors of training response.

## 1. Introduction

Walking ability is a key determinant of independence, quality of life, and survival, and it significantly impacts survivors with severe walking impairment [1]. Defined as a self-selected walking speed of less than 0.4 m/s [2], stroke survivors with severe waking impairment represent approximately 20 percent of survivors [3,4]. Those meeting this classification are more likely to be homebound, sedentary, experience fear of falling, and are at greater risk of imbalance and falls than survivors with less impairment [3].

Limitations and challenges of rehabilitation in this group are multifaceted and most commonly related to the severity of physical impairments such as muscle weakness, spasticity, pain, imbalance, and poor exercise tolerance [5,6]. Current clinical practice guidelines provide general guidance on the best evidence-based walking rehabilitation strategies to improve walking capacity and balance after stroke [7]; however, less is known about the most effective strategies to improve walking outcomes in chronic stroke survivors with severe walking impairment.

As this subgroup is often underrepresented in walking rehabilitation trials, little is known about how well this subgroup responds to intensive walking rehabilitation strategies, and the few studies available suggest limited responsiveness [1,8,9]. While this knowledge is helpful from the perspective of optimizing clinical trial design and interpretation [10,11,12], excluding or not customizing training protocols to meet the needs of this subgroup may lead to an unintended consequence and further widen the gap in knowledge on the most effective interventions to optimize outcomes for all stroke survivors. Therefore, more walking rehabilitation studies targeting chronic stroke survivors with severe walking impairment are needed to obtain greater insight into novel approaches to improve walking capacity and associated outcomes in this population.

Backward locomotor treadmill training (BLTT) is a novel non-body weight-supported walking rehabilitation protocol that is well tolerated across stroke severity types [13,14]. While its underlying mechanisms remain under investigation, as a concept, backward walking (BW) has potential attributes that make it a promising rehabilitative strategy. For example, compared to forward walking, BW has been suggested to require greater focus and induce greater cerebral activation than walking in the forward direction and may better engage cerebral pathways damaged by stroke [15,16,17]. Moreover, BW is performed in the absence of peripheral visual feedback and may, therefore, optimize sensory system integration and postural stability by activating complementary sensory processing pathways, such as the somatosensory and vestibular pathways [18]. From a musculoskeletal physiology perspective, BW has been reported to activate key postural muscles, such as the trunk, hip, and knee muscles, more than forward training [18,19,20,21,22]. Also, given that BW requires concurrent hip extension and knee flexion to bring the leg posterior to the trunk, repetitive backward locomotion may work to reverse the maladaptive flexor–synergy pattern characteristic of post-stroke gait [18]. Specific to the BLTT protocol, the primary use of a treadmill in lieu of overground training provides a controlled training environment that allows for more practice steps and faster, more consistent, and adjustable training speeds over time [23,24]. Lastly, the absence of a bodyweight support harness may encourage greater weight bearing on the paretic leg [25,26,27], which can lead to improved strength and independent weight bearing and walking spatiotemporal characteristics over time.

Previously, our group tested BLTT in a cohort of chronic stroke survivors and found it to be safe and beneficial across impairment levels following six training sessions [13,14]. However, while subjects with mild (SS walking speed ≥ 0.8 m/s) and moderate (SS walking speed 0.4–0.8 m/s) symptoms experienced clinically meaningful improvements (MCID-0.16 m/s) [28], the outcome was not as robust in the subgroup with severe walking impairment [13]. Past reports have suggested that stroke survivors with severe motor impairments may require more time and training to see improvement compared to individuals with less severe impairment [3,9,29,30]. Therefore, the primary objective of this pilot randomized controlled study is to explore the impact of extended training on the achievement of the MCID in walking speed. Also, to better understand the impact of BLTT compared to a conventional yet equally rigorous walking rehabilitation approach, forward locomotor treadmill training (FLTT), a subtype of task-specific training, was chosen as the most suitable control. The working hypothesis is that BLTT will lead to greater and longer-lasting improvement in walking speed, spatiotemporal symmetry, and dynamic postural stability.

## 2. Materials and Methods

### 2.1. Setting and Participants

This study was approved by the University of Cincinnati Institutional Review and was performed in the Neurorecovery Lab from October 2020 to December 2023 and preregistered at clinicaltrials.gov (NCT04721860) prior to commencing enrollment. Study participants were greater than six months post-stroke (chronic) and recruited from the community. According to the Declaration of Helsinki recommendations, all participants provided written informed consent before enrollment. Inclusion and exclusion criteria were 18–80 years of age, residual walking impairment secondary to ischemic/hemorrhagic stroke, walking speed less than or equal to 0.4 m/s, ambulating at least 10 m without a walker (other walking aides were admissible), and ability to maintain greater than or equal to 0.3 mph speed for a 3 to 6 min interval on the treadmill. In addition, all participants were asked to abstain from formal physiotherapy throughout the four-week intervention period (three sessions/week). Exclusion criteria included severe lower extremity spasticity (modified Ashworth > 2/4), an adverse health condition that might affect walking training (severe: arthritis, cardiopulmonary disease, ataxia, or hemineglect), significant language barrier that may interfere with the ability to follow instructions during training and testing, and untreated depression [>10 on the Patient Health Questionnaire (PHQ9)]. Due to the COVID-19 pandemic, all participants were required to wear a surgical mask during training and outcome testing, per institutional requirements. Participants meeting the inclusion and exclusion criteria were randomized to either BLTT or FLTT before the initiation of training (day 1) using a REDCap algorithm [permutated block randomization with stratification factor (average self-selected walking speed < 0.21 m/s vs. 0.21–0.4 m/s)] generated and concealed by the lead statistician, H.J.S., as shown in Figure 1.

### 2.2. Training Protocol

Subjects underwent twelve 30 min sessions of BLTT or FLTT over four weeks. As previously described [14], participants were connected to a safety harness without bodyweight support during training as a safety precaution and faced either the back (BLTT) or the front (FLTT) of the treadmill. In addition, as a safety and tolerability measure, all participants were required to manually hold on to the ipsilesional (less affected side) Biodex Gait Trainer 3 (Biodex, Shirley, NY, USA) treadmill handrail continuously during training and provided standardization across sessions and between groups. The treadmill belt speed started at the most comfortable training speed established during the screening visit. The speed was adjustable throughout the training period based on patient preference or per therapist recommendation, barring safety concerns. Additionally, to facilitate safe stepping with training, therapists were allowed to provide physical assistance to initiate stepping on the paretic side during training (Range: Contact Guard to Max Assist), and the amount of support was tracked throughout the twelve training sessions, along with other metrics, such as training speed, step count, and step length (paretic and non-paretic), per training session; see Appendix A for training data. Per protocol, two-minute rest breaks were given after each bout of six minutes of treadmill training (e.g., 6 min-2 min-6 min-2 min-6 min-2 min-6 min), and they were allowed to take additional rest breaks in between intervals if needed. On subsequent training days (#2–12), the starting treadmill speed was set to the last treadmill speed reached on the preceding training day and was adjustable throughout the duration of training. The treadmills were run at 0% incline. A short 5 min overground (forward walking) period followed each training session for both groups to allow for cooldown and to promote translation to overground walking.

### 2.3. Outcome Measures

#### 2.3.1. Overground Walking Speed (10-Meter Walk Test)

The 10-meter walk test (10 MWT) is the gold standard measure of post-stroke walking function that reflects overall mobility and health status [31]. A trained physical therapist used a handheld stopwatch to time participants over the 10-meter walkway to obtain the walking speeds. A two-meter space was provided prior to the starting and ending markers to allow for acceleration and deceleration, respectively. The timing started when the participant’s lead leg broke the plane of the starting marker and stopped when both legs crossed the marker at the end of the path. Subjects were allowed to use their home assistive device and orthosis. They were asked to walk at their fastest speed without running. Two 10 MWT fast trials were performed and averaged for analysis. Blinded assessments were performed at baseline, 24 h, 30, and 90 days POST (following twelve training sessions) by a trained accessor unaware of group allocation or study objectives). Furthermore, to track daily individual changes in walking speed, the treatment therapists (unblinded to the intervention but unaware of the study objectives) performed the 10 MWT (two trials) prior to the start of each training session. The primary outcome was the proportion of participants achieving MCID by 24 h POST. The longer-term effects of training participants returned for testing at 30 and 90 days post-training were evaluated. Lastly, to illustrate the relative change in walking speed as a proportion of baseline, the percentage change in walking speed (PRE-POST) was determined for all time points.

#### 2.3.2. Symmetry During Overground Walking

Spatial (step length) and temporal (% single support time) asymmetry is common after stroke and is associated with paretic limb dysfunction, dynamic postural instability, and increased likelihood of falls [32,33]. Step length and single support times were collected simultaneously as the 10 MWT trials using the ProtoKinetics ZenoTM Walkway Gait Analysis System (ProtoKinetics LLC, Havertown, PA, USA) and analyzed by study personnel (S.N.) blinded to group allocation. The following equation was used for measuring spatial and temporal symmetry during the 10 MWT: [1 − |Paretic − Non-paretic|/(Paretic + Non-paretic)] × 100%, where the integer values for the respective measure based on the limb (i.e., paretic or non-paretic) are inputted. Possible symmetry index values range from 0 to 100%, where 100% means perfect symmetry and 0% means complete asymmetry [34].

#### 2.3.3. Dynamic Balance—Instrumented Timed up and Go

The 3-meter timed up and go test (TUG) is a commonly referenced and reliable time-based assessment of dynamic postural stability in stroke survivors during daily functional movement tasks [35,36]. Subjects were instructed to sit with their back against the chair (seat height 46 cm, arm height 67 cm) and on the word “go”, stand up, walk at a comfortable speed past the 3 m mark, turn around, walk back, and sit down in the chair. Two trials are averaged and documented in seconds. A blinded assessment was performed at baseline, 24 h, 30, and 90 days POST and was consistently performed following the completion of the 10 MWT trials. Due to a lack of studies in the literature specific to response thresholds for chronic stroke survivors with severe walking impairments for the TUG test, established thresholds for chronic stroke survivors with milder impairments were used as a reference (minimal detectable change = −2.9 s; smallest real difference (SRD) of 23%) [37].

#### 2.3.4. Statistics

The study sample size was based on the planned comparison of the BLTT group to the FLTT group for the change in the 10 MWT from baseline to 24 h POST. Since no previous backward treadmill training studies specific to chronic stroke survivors with severe walking impairment were available, an early study by Yang et al., 2005 [38] comparing backward walking and conventional training (with forward task-specific training) was used for sample size determination. It was estimated that a sample size of 9 subjects per group would achieve 80% power to detect a mean difference in change in the 10 MWT of 0.13 m/s between groups, with an estimated standard deviation (SD) of 0.11 m/s and a significance level of 0.10.

Statistical analyses were performed using GraphPad Prism version 10.3.0. Given the small sample size and non-normally distributed nature of some outcome measures (e.g., 10 MWT, 3M-TUG), the Mann–Whitney U test was used for all continuous outcome measures, and Fisher’s exact tests were used for categorical measures. Within-group analyses were performed with the Wilcoxon matched pairs signed rank test. The median and interquartile range (1st–3rd) were provided throughout the manuscript unless otherwise specified. Beyond the comparison change in walking speed between groups at 24 h POST, alpha was set at 0.05 for all significance testing. Given the exploratory nature of this pilot, statistical analyses did not correct for multiple comparisons.

## 3. Results

In total, a convenience sample of 22 individuals was screened, as shown in Figure 2. Eighteen participants who met the screening inclusion and exclusion criteria were randomized. To maintain study integrity and protocol fidelity, three participants (BLTT: 1/FLTT: 2), who initially passed screening and were randomized, no longer qualified for this study because their self-selected walking speed exceeded the 0.4 m/s cutoff prior to training initiation. Additionally, one participant in the BLTT group dropped out after six sessions due to logistical issues with getting to the lab for training and was no longer able to participate. Of those screened, 14 (BLTT: 7; FLTT: 7) completed the study protocol. Baseline demographics and outcome measures were comparable between groups, as shown in Table 1.

### 3.1. Safety/Tolerability/Study Fidelity

Ninety-nine percent (166/168) of the ascribed training sessions were attended. The training was well tolerated, whether backward or forward, with no serious adverse events, including cardiac, cerebrovascular, or orthopedic injuries, and no emergency room visits, hospitalizations, or fatalities. See Appendix A for subject-level details. At the beginning of training, several participants (BLTT = 7/FLTT = 4) required therapy assistance for paretic step initiation (contact guard—max assist); however, less to no assistance was needed by the 12th training session. The median (IQR) cumulative steps/subjects taken over 12 sessions were 15,117 (13,690–16,252) for BLTT and 18,842 (13,397–23,047) for FLTT, as shown in Appendix A.

### 3.2. Overground Walking Speed at 24 h POST (Primary Outcome)

Following 12 training sessions (24 h POST), 29% (2/7) of the subjects in the BLTT and 14% (1/7) FLTT groups achieved the MCID, as shown in Figure 3.

At 24 h POST, the median (IQR) percent change in walking speed from baseline was 28.9 (9.01–36.7) with BLTT and 17.4 (12.6–39.7) with FLTT, as shown in Figure 4. There was no statistically significant within-group change with BLTT (z = −1.76, *p* = 0.08). In contrast, FLTT resulted in with-group improvement (z = −2.42 *p* < 0.02). However, no significant between-group differences were seen (*p* = 0.80), as shown in Table 2.

### 3.3. Retention of Walking Speed

On day 30 POST, 71% (5/7) in BLTT and 86% (6/7) in FLTT maintained walking speeds equal to or faster than baseline (PRE). The percent change in walking speed was 21.3 (−0.03–57.6) for BLTT (z = −1.35, *p* = 0.22) and 19.1 (12.1–38.4), z = −2.15, and *p* = 0.03; however, there were no significant differences between groups; *p* > 0.99. On day 90 POST, 58% (4/7) in BLTT and 86% (6/7) in FLTT maintained walking speeds faster than baseline. The percent change in walking speed was 29.0 (−0.56–49.75) for BLTT (z = −1.23, *p* = 0.22) and 23.0 (11.3–40.3) for FLTT (z = −2.15, *p* = 0.0.3); however, there were no significant differences between groups; *p* > 0.99.

### 3.4. Spatial (Step Length) Symmetry Index

The median (IOR) change in step length symmetry at 24 h POST was 3.17 (−2.95–4.87), z = −0.72, and *p* = 0.47, signifying an improvement, but it trended downward for FLTT −1.61 (−3.19–3.39), z = −0.08, and *p* = 0.94. Similar patterns were seen on days 30 and 90 of follow up; however, there were no within or between-group differences at all time points.

### 3.5. Temporal (% Single Support Time) Symmetry Index

The median (IOR) change in % single support time symmetry at 24 h POST was 2.75 (1.17–4.19), z = −1.60, and *p* = 0.11 for BLTT and 2.36 (−4.81–4.02), z = −0.08, and *p* = 0.94 for FLTT. The improvement in the temporal symmetry index regressed by days 30 and 90 of follow up for both groups, with no within- or between-group differences at all time points.

### 3.6. 3-Meter Timed Up and Go

The median (IOR) improvement in 3M TUG completion time at 24 h POST was 9.88 s (3.04–24.8), z = −1.42, and *p* = 0.16 for BLTT and 6.59 s (2.90–9.21), z = −1.76, and *p* = 0.08 for FLTT, as shown in Figure 5. On days 30 and 90 of follow up, there were no significant within-group differences for BLTT relative to baseline. Conversely, FLTT resulted in within-group improvement (day 30: z = −2.42, *p* = 0.01; day 90: z = −2.15, *p* = 0.03). However, there were no differences between groups at all time points. Using the SRD as a reference [37], 43% in BLTT (3:7) and 29% in FLTT (2:7) exceeded the 23% threshold.

## 4. Discussion

To our knowledge, this is the first walking rehabilitation trial centered exclusively on chronic stroke survivors with severe walking impairment. The primary objective of this pilot study was to test whether twelve sessions of non-bodyweight-supported backward treadmill training could lead to clinically meaningful differences in walking speed (MCID) in this understudied subgroup. In contrast to our working hypothesis, the majority of subjects did not achieve the minimal clinically important difference (MCID) in either the active (BLTT) or control group (FLTT). The subject-level results demonstrated high variability in training response, with 21% achieving MCID (2 BLTT:1 FLTT), 64% achieving progressive improvement from baseline (3 BLTT: 6 FLTT), and 14% performing worse than baseline (2 BLTT: 0 FLTT) at the primary endpoint (24 h POST), as shown in Figure 3.

Limited information is available about backward treadmill training in the population with severe walking impairment. However, in the context of past walking rehabilitation studies using forward treadmill training with and without bodyweight support, the mean change in baseline walking speed following twelve training sessions was comparable to historical controls in the literature [+0.06 m/s (95% CI 0.03–0.09), 47 trials, N = 2323) [39], many of which had many more training sessions. Specific to chronic stroke survivors with severe walking impairment, secondary analysis of the AMBULTAE trial reported a within-group (N = 11) change of 0.9 m/s (SD = 0.9) following forty-eight 30 min forward treadmill training sessions performed over 16 weeks [9]. Therefore, future studies with longer training sessions than were feasible in our present study may be instructive on the benefits of extended training and provide greater context for interpreting training effects in relation to past studies.

### 4.1. Backward vs. Forward Training on Spatiotemporal Measures

Additionally, this study aimed to characterize the differences in immediate and longer-term effects of backward versus forward treadmill training on spatiotemporal walking characteristics. Anchored by previous reports that used variations of backward walking strategies, in the aging population [40] with multiple sclerosis [41] and stroke [13,22,42], it was hypothesized that BLTT would be advantageous in improving walking speed and spatial and temporal symmetry compared to FLTT. While this aim was not powered to detect significant differences between groups definitively, we anticipated that BLTT would demonstrate a clear trend of outperforming FLTT across outcome measures; however, the results did not meet our working hypothesis. Of note, although point estimates suggest that subjects in the BLTT group had a higher magnitude percent change in walking speed among responders, this group also had two poor responders (BLTT #1 and 2), which confounded the results. In contrast, those in the FLTT group had more consistent results across time points. One possible explanation is that FLTT is a more familiar and task-specific training strategy, enabling subjects to train at progressively faster speeds and with less assistance, as shown in Appendix A. With this said, although not at a level of significance, subjects that underwent BLTT trended toward resulting in improved step length symmetry (particularly at 24 h and 30 days POST) compared with FLTT, which demonstrated worsened symmetry post-training, as shown in Table 2.

This observation is consistent with past rehabilitation studies reporting the limited impact of task-specific forward walking on symmetry [43,44]. In contrast, the improvement in spatial symmetry with BLTT may be related to the longer stance times, which may encourage greater weight bearing on the paretic limb during backward extension of the non-paretic limb. Although lower extremity (i.e., ankle) strength and propulsion forces were not measured in this study, the reduction in therapy assistance and increased non-paretic step length during training suggest that BLTT may lead to greater paretic limb weight-bearing capacity and improvement in step length symmetry, as shown in Appendix A. Interestingly, although step length asymmetry and stance time are related, neither BLTT nor FLTT resulted in a sustained change in % SST symmetry. This finding may have resulted from the primary use of a treadmill instead of overground training, which may not have allowed for sufficient targeting of stance time asymmetry, which relies heavily on sensory feedback from hip flexor afferents, limb load receptors, and cutaneous feedback, which is more achievable overground than on a continuously moving platform [44,45,46,47]. Therefore, future protocols targeting temporal symmetry may benefit from the addition of extended overground training. Alternatively, bidirectional treadmill training paired with overground training may provide this subgroup with a more comprehensive training strategy.

### 4.2. Backward vs. Forward Training on Dynamic Balance

Another objective of this study was to determine the impact of BLTT compared to FLTT dynamic balance, as measured by completion time on the 3M TUG test [48]. Based on the notion that backward walking better targets sensorineural integration and postural control and supported by past studies in the elderly [40] and neurologic disorders [41,49,50], the working hypothesis was that BLTT would reduce task completion time. Although we found no significant between-group differences, the magnitude of improvement in responders favored the BLTT group, as shown in Figure 5. However, as with the 10 MWT, there was more variability in the response with BLTT than FLTT.

Of note, although the 3M TUG test is widely used in stroke rehabilitation as a predictor of falls, the MCID and SRD referenced in this study were derived from chronic stroke survivors with milder impairment (SS walking speed ≥ 0.8 m/s) [37], and such references were unavailable for the severely impaired. Therefore, both threshold references were incorporated in this study to provide a transparent assessment of the influence of each training paradigm on dynamic postural stability based on the best available evidence. Nevertheless, reliability studies assessing dynamic postural stability outcomes in stroke survivors with severe motor impairments are needed to establish more appropriate thresholds for characterizing the impact of interventions in this subgroup.

### 4.3. Factors Predicting Response

The determinants of responsiveness to rehabilitation interventions remain elusive; however, factors such as sociodemographics (i.e., age, sex, race, socioeconomic, education, social support), genetics, lesion size, location, chronicity, comorbidities, and activity level have been reported to play a role [1,30,51,52,53]. Recently, a baseline self-selected walking speed of <0.4 m per second is gaining traction as a key determinant of walking capacity and postural stability outcomes [1,54]. However, in this study, where baseline self-selected speeds ranged between 0.11 and 0.39, no clear relationship between walking speed and training response was observed. For example, the two subjects in the BLTT group with the “fastest” baseline self-selected walking speeds (BLTT #1 and 2) had the poorest response in walking speed, as shown in Figure 4A. Although the ability to make strong conclusions on the determinants of response was limited by its relatively small sample size, response to training could not be explained by age, stroke chronicity, type, location, the use of walking aid or orthosis, training intensity, or medications. While speculative, one commonality that was seen with poor responders at 24 h POST was the level of reported fatigue (≥5 out of 10) on the post-training questionnaire (Appendix A). Although those subjects (BLTT #1 and 2) completed all training sessions, it is possible that the training intensity may have been more than these subjects were accustomed to performing (i.e., sedentary lifestyle). An alternative explanation is that progressive backward training may have altered their post-stroke asymmetric adaptive gait pattern, resulting in an improvement in spatial symmetry but diminishing gait speed, as shown in Appendix A. Lastly, we found no obvious relationship between interval weekly physical activity level and retention of walking speed or TUG completion time on days 30 and 90 of follow up, as shown in Appendix A.

### 4.4. Key Logistical Considerations

A key observation from this study was all participants in our study were unable to drive or ambulate long distances (e.g., university garage to lab) independently compared to survivors with lesser motor impairment; therefore, significant logistical planning and provision of travel funds and personnel support are needed to successfully enroll and retain study participants in this population [29,55].

### 4.5. Limitations

This study had limitations that may impact study interpretations and generalizability. Firstly, based on a convivence sample estimation and limited pilot funding, this study was underpowered and lacked sufficient statistical power to delineate differences between BLTT and FLTT. Indeed, a larger sample size may have helped to reduce the impact of within-subject variability on study interpretation and may also have provided greater power for gaining preliminary insight into the predictors of response. Also, the number of training sessions used in this study was smaller than in contemporary rehabilitation studies, which may have influenced outcomes as it may not have accurately demonstrated the full potential of our training approach. Nevertheless, our preliminary results can assist in designing future well-powered, hypothesis-driven, and higher-dosed studies for this subgroup. Moreover, although by chance, all subjects in the BLTT group used a quad cane at baseline, the use of assistive devices was more distributed in the FLTT. However, since subjects underwent stratified randomization based on their baseline self-selected walking speed, were required to use their assistive devices during all testing, and had no clear relationship between the use of walking aid and spatiotemporal measures, there is a low likelihood that this factor influenced the study results or their interpretation. Regarding the impact of training on dynamic postural stability, due to the unavailability of references in the severe population, the thresholds for 3M TUG, based on stroke survivors with mild walking impairment, were used and should be interpreted in this context. Lastly, while all participants in this study analysis were in the severe walking category, non-ambulators were excluded; therefore, our findings are not generalizable to this population. A modified study protocol, including bodyweight support with graduated weaning, may be needed to facilitate training in future studies.

## 5. Conclusions

In this preliminary study involving chronic stroke survivors with severe walking impairment, twelve sessions of BLTT did not lead to the group-level achievement of the MCID in walking speed. Additionally, there were no significant post-training differences between BLTT and FLTT in walking speed, spatiotemporal symmetry, or dynamic postural stability. Future well-powered dosing trials and mechanistically driven studies are needed to optimize and identify predictors of training response.

## Figures and Tables

**Figure 1 brainsci-15-00437-f001:**
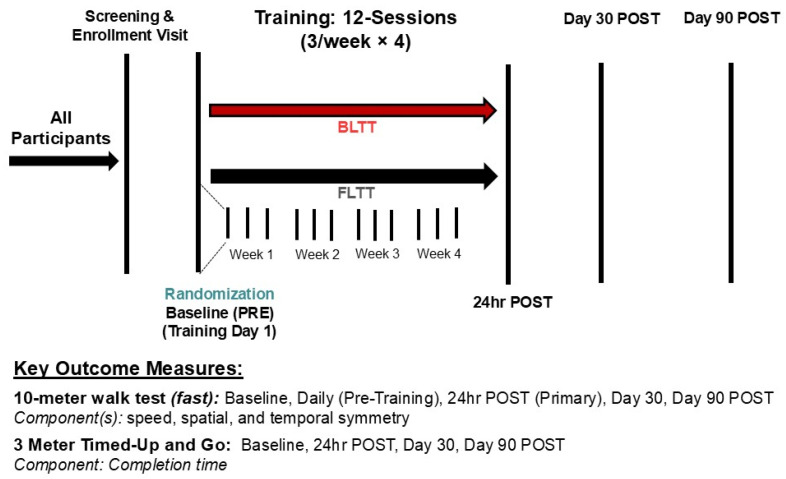
Study design.

**Figure 2 brainsci-15-00437-f002:**
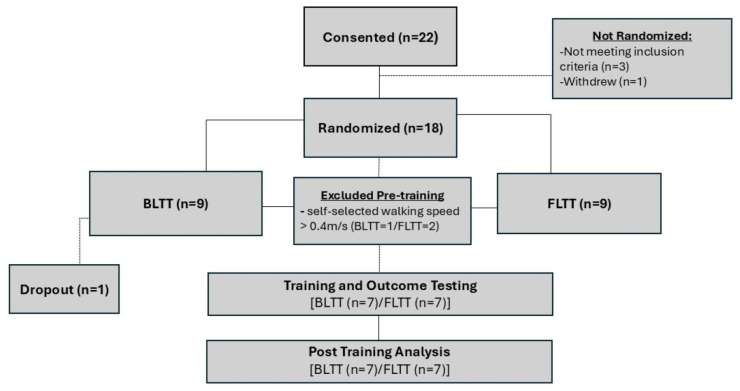
Study profile.

**Figure 3 brainsci-15-00437-f003:**
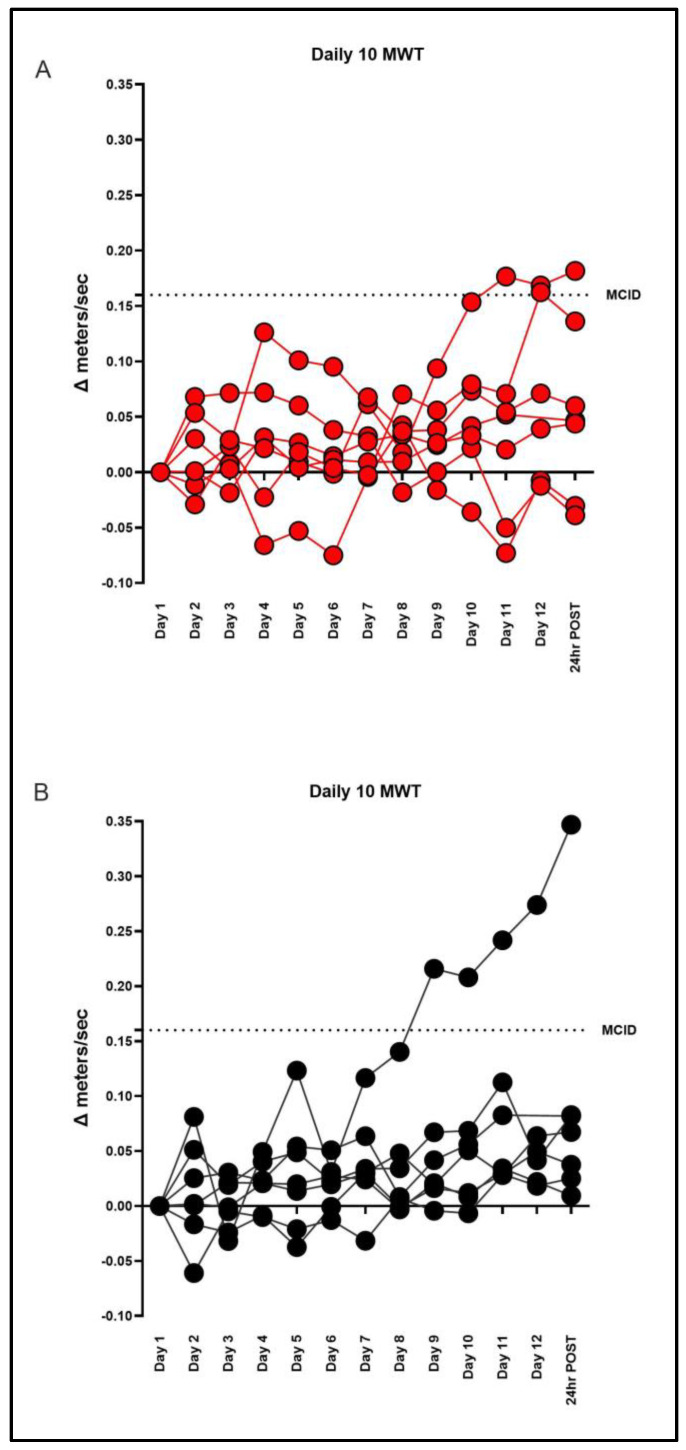
Daily changes in individual walking speed with backward (BLTT) (**A**) and forward (FLTT) locomotor treadmill training (**B**); 24 h POST represents walking speed following twelve training sessions. The dotted line represents the clinically meaningful important difference (MCID) for the 10-meter walk test [10 MWT (0.16 m/s)] [28].

**Figure 4 brainsci-15-00437-f004:**
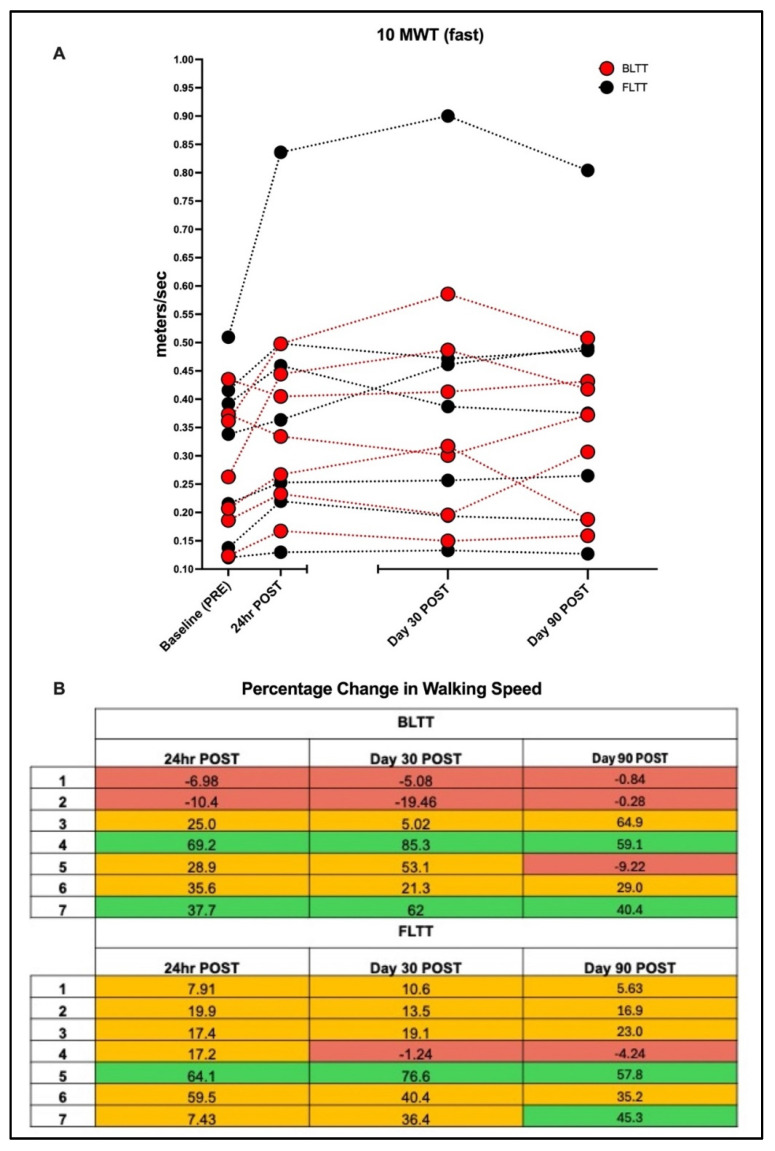
Individual walking speed on the 10-meter walk test (10 MWT) at baseline (pre-training), 24 h, 30, and 90-days post-training (**A**). Backward locomotor treadmill training (BLTT); forward locomotor treadmill training (FLTT). Percentage change in walking speed relative to baseline (**B**). Green boxes indicate trials where the subject achieved or exceeded the minimal clinically important difference for the 10 MWT [28], some improvement from baseline (orange), or worsened to no change from baseline (red).

**Figure 5 brainsci-15-00437-f005:**
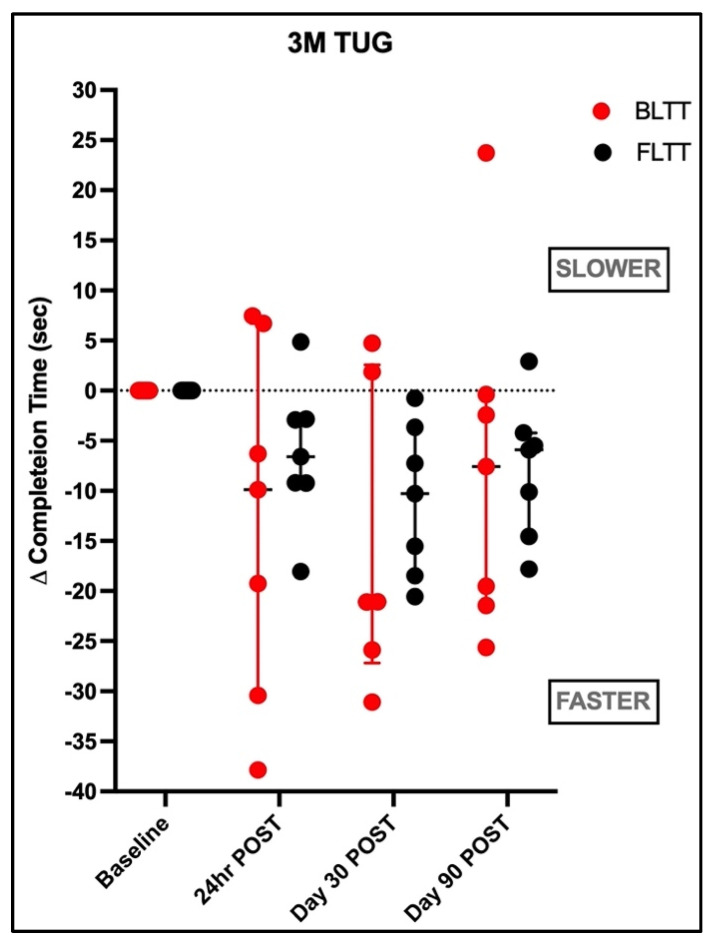
Performance on 3-meter timed up and go.

**Table 1 brainsci-15-00437-t001:** Clinical characteristics of the participants.

Category	Study Groups		*p*-Value
	BLTT (*n* = 7)	FLTT (*n* = 7)	
Time since Stroke (mo)	14.0 (7.00–24.0)	25.0 (10.0–47.0)	0.32
Age (yrs)	57.7 ± 7.50	53.9 ± 7.41	0.47
Mini Mental Status Exam	29 (28–30)	30 (29–30)	0.36
Patient Health Questionnaire-9	3 (2.75–6.00)	3 (1.50–5.00)	0.83
Sex			1.00
Male	5 (71.4%)	6 (85.71%)	
Female	2 (28.6%)	1 (14.29%)	
Assistive Device			0.10
None	0 (0.00%)	0 (0.00%)	
Cane	0 (0.00%)	2 (28.6%)	
Quad Cane	7 (100%)	4 (57.1%)	
Hemiwalker	0 (0.00%)	1 (14.3%)	
Orthotic Device (AFO)			1.00
Yes	3 (42.9%)	4 (57.1%)	
No	4 (57.1%)	3 (42.9%)	
Stroke Type			1.00
Ischemic	4 (57.1%)	4 (57.1%)	
Hemorrhagic	3 (42.9%)	3 (42.9%)	
Lateralization			0.59
Right	3 (42.9%)	5 (71.4%)	
Left	4 (57.1%)	2 (28.6%)	
Localization			0.46
Hemispheric	7 (100%)	5 (71.43%)	
Cerebellar	0 (0%)	2 (28.57%)	
10 MWT (m/s)			
Self-selected (SS)	0.20 (0.17–0.30)	0.26 (0.15–0.30)	1.00
Fast (FP)	0.26 (0.20–0.37)	0.34 (0.18–0.40)	1.00
Step Length Symmetry	85.7 (74.6–88.1)	81.6 (46.1–97.2)	0.90
% Single Support Time	83.9 (68.1–85.5)	84.2 (76.6–92.3)	0.32
3-Meter Timed Up and Go	47.1 (44.9–72.9)	44.0 (34.6–69.4)	0.40

Demographic variables and baseline comparison of outcomes between treatment groups. Values reported median (IQR), except for age, represented as mean ± SD.

**Table 2 brainsci-15-00437-t002:** Spatiotemporal characteristics and dynamic balance following twelve training sessions.

Outcome	Group	24 h POST	*p*-Value	Day 30 POST	*p*-Value	Day 90 POST	*p*-Value
Δ 10 MWT (m/s)	BLTT	0.05 (0.01–0.08)	0.80	0.03 (−0.01–0.14)	>0.99	0.04 (−0.00–0.13)	>0.99
	FLTT	0.07 (0.03–0.08)		0.06 (0.03–0.07)		0.05 (0.03–0.09)	
Δ SLS	BLTT	3.17 (−2.95–4.87)	0.54	3.79 (1.58–5.95)	0.26	1.09 (−6.05–6.93)	0.18
	FLTT	−1.61 (−3.19–3.39)		−0.90 (−1.60–2.24)		−4.54 (−7.14–−1.12)	
Δ %SST	BLTT	2.75 (1.17–4.19)	0.62	−2.10 (−1.60–2.24)	0.71	0.60 (−0.86–2.34)	0.10
	FLTT	2.36 (−4.81–4.02)		−1.77 (−5.42–1.49)		−1.85 (−3.69–−0.51)	
Δ 3M TUG (s)	BLTT	9.88 (3.04–24.8)	0.53	21.1 (15.3–28.5)	0.37	7.59 (1.91–20.5)	0.80
	FLTT	6.59 (2.90–9.21)		10.3 (6.36–17.0)		5.9 (5.17–12.3)	

Numbers are median and interquartile range in parenthesis. Change in step length symmetry (Δ SLS indicates) relative to baseline; change in percent single support time symmetry (Δ %SST) during the 10-meter walk test relative to baseline; change in completion time on the 3-meter timed up and go test (Δ 3M TUG) relative to baseline.

## Data Availability

The original contributions presented in this study are included in the article/Appendix A. Further inquiries can be directed to the corresponding authors.

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
