# Peer review of "Exploring the Impact of Backward and Forward Locomotor Treadmill Training in Chronic Stroke Survivors with Severe Post-Stroke Walking Impairment: A Single-Center Pilot Randomized Controlled Trial"

_brainsci, 2025, doi:10.3390/brainsci15050437_

Round 1

Reviewer 1 Report

Comments and Suggestions for Authors

The manuscript shows the impact of a Treadmill Training in Chronic Stroke Survivors with Severe Post-Stroke Walking Impairment. I have some considerations for you:

Review the journal's citation guidelines. References in the text should be enclosed in square brackets.

Lines 97-99: not part of the stated objectives or hypotheses. Move this information to the methodology section.

Were other comorbidities or treatments considered in the inclusion and exclusion criteria?

Line 118: the total number of participants belongs in the results section, in the sample section the sample type in general (stroke survivors for example) should be reflected. Same comment on line 120.

Figure 1 are results of the final sample.

The figure should be cited in the text.

Consider adding a figure with the timeline of the training protocol.

It would be advisable to add the reliability and reproducibility of the tests used for this population.

The procedure should be fully explained in order to be reproducible: how many investigators? Where was the study conducted? Was the 10MWT the first test or was there another order to follow?

Statistical analysis:

Again, the explanation is general without defining the total number of participants, which should be in the results.

Results:

A picture of both treatments would also be advisable and not only as supplementary material.

Abbreviations must be explained at the table footnote.

Line 270: there is a superscript reference.

Line 373: revise ‘convivence’.

Discussion: It would be useful to organize the information in relation to the proposed objectives and explain a little more about the sample used. I understand that there is a previous study, but the information cited in the present study remains scarce. The same is true for both types of surveys.

References:

Please follow the journal instructions

Reviewer 2 Report

Comments and Suggestions for Authors

General comment

This study addresses an important gap in stroke rehabilitation by focusing on chronic stroke survivors with severe walking impairment, a population often underrepresented in research. The comparison between backward (BLTT) and forward (FLTT) locomotor treadmill training adds novelty, as does the exploration of dosing effects (12 sessions vs. previously tested 6 sessions). However, the lack of significant between-group differences and the small sample size limits the novelty of the findings.

The manuscript is well-structured and clearly written. The objectives, methods, and results are presented logically. Nonetheless, some sections could benefit from more concise language, and the discussion could better highlight the clinical implications of the findings.

Specific Comments

Abstract

  • Pg 13; Ln 35-37: The conclusion states that “severely impaired chronic stroke survivors can achieve MCID in walking speed when offered additional training” despite only 21% of participants achieving this benchmark, which is an overgeneralization.

Introduction

  • Pg 2; Ln 43-59: The introduction provides a good rationale for studying severe walking impairment but could better emphasize why this subgroup is particularly challenging to rehabilitate. More specific data on the lack of effective interventions for this population would strengthen the argument.
  • Pg 3; Ln 63-62: The hypothesis about BLTT’s advantages (e.g., sensorimotor integration) is well-articulated, but the rationale for comparing it to FLTT is less clear. Why is FLTT the appropriate comparator, given its familiarity?
  • Pg 2, Ln 91-94: the rationale for doubling the training sessions from 6 to 12 is not adequately supported by previous literature. The authors should provide stronger evidence-based justification for why 12 sessions might be the optimal dose.
  • There is no explicit hypothesis statement regarding the expected difference between BLTT and FLTT.

Methods

  • Pg 3; Ln 121-128: the description of the randomization process using the “REDCap algorithm” lacks sufficient detail. The authors should specify the type of randomization (e.g., block, stratified) and how the allocation concealment was maintained.
  • Pg 4; Ln 130-151: Once again, the training protocol is described thoroughly, but the rationale for the 12-session dose (beyond “doubling” the previous 6 sessions) is not well justified. Is there evidence supporting this dose?
  • Pg 4; Ln 130-: The training protocol mentions adjustable treadmill speeds “based on patient preference or therapist recommendation,” but there is no quantification of how often speeds were adjusted or the range of speed used.
  • The “therapy assistance” (e.g., contact guard to max assist) is not standardized or quantified across sessions.
  • The rationale for 6-minute training / 2-minute rest interval is not provided.
  • Pg 4: Ln 130-135: Clarify why participants held the handrail continuously. Could this confound the results by reducing balance challenges?
  • Pg 4; Ln 153: The timing of assessments (24hr, 30-, 90-days post) regarding the 10MWT could be better justified. Why these intervals?
  • Pg 5; Ln 183: The 3m TUG is a good choice, but the reliability in severe stroke populations should be discussed, given the cited MCID is for milder impairment.
  • Pg 5, Ln 193-201: The statistical section does not include a proper power analysis to justify the sample size. Given the small sample (n=7 per group), the authors should acknowledge this limitation, even though the design is a small-scale (pilot) study.

Results

  • Pg 5; Ln 204: Table 1 shows baseline characteristics but does not include all relevant factors that might influence outcomes (e.g., previous rehabilitation experience, medication use).
  • Pg 5; Ln 206-212: the authors mentioned “no serious adverse events”, but they did not define what constitutes serious and non-serious events, nor did they systematically report all adverse effects.
  • Pg 6; Ln 227-233. The p values are reported for some comparisons but not others, making interpretation difficult.
  • Pg 6, Ln 227-233: the primary outcome (MCID achievement) is reported clearly, but the lack of statistical significance between groups is downplayed. The discussion should address this more critically.
  • Pg 7; Ln 282-287: Despite highlighting the variability in response (21% achieving MCID), there is insufficient analysis of factors that might predict or explain this variability, which is crucial for clinical application.

Discussion

  • Pg 9; Ln 280-287: the authors make a strong claim about the effectiveness of additional training despite the small sample size and limited statistical significance, without adequately acknowledging these limitations.
  • Pg 10; Ln 289-295: despite being a comparative study, the discussion does not thoroughly analyze why there were no significant differences between BLTT and FLTT, which is presumably a key research question.
  • Pg 10; Ln 291-295: the discussion does not comprehensively compare the findings with other studies on treadmill training in severe stroke, limiting the contextualization of the results.
  • Pg 10-11: How the findings could be implemented in clinical practice needs more attention. This should include considerations of resource requirements and feasibility.
  • Pg 11; Ln 354-357: The exploration of predictors (e.g., fatigue) is insightful but speculative due to the small sample. This could be framed more cautiously.

Reviewer 3 Report

Comments and Suggestions for Authors

Based on Author's previous research, this study was to determine if doubling the number of training sessions could result in survivors achieving MCID in walking speed, capacity, and dynamic postural stability. However, the research method is to compare whether there are differences between forward treadmill training (FLTT) and backward treadmill training (BLTT). In this single-center, assessor-blinded, randomized controlled pilot trial, some between-group differences suggested potential for improvement in gait, but few significant differences were found between FLTT and BLTT training. However, the author did not use data from past treatments with fewer times for comparison, which resulted in the final conclusion of the article being unable to be confirmed from the results of this article.

Major issue: 

As mentioned above, the biggest problem is that the conclusion does not match the research design. If the authors wants to know whether more treatments can achieve the desired effect, he/she should analyze and compare the effects of different treatment times in the same article, and the two comparisons should have similar evaluation time points.

Minor issue:

1. line 110 and table 1 : One group obviously all used with quadricane, while the other group used a variety of walking aids. Although the two groups did not seem to be "significantly different" based on the small case number analysis, such a comparison is actually significantly different. The author should analyze/explant whether different walking aids interact with the training method.

2. Methods : Although this is a pilot study, the authors should explain how the case number is determined.

3. Figure 2 and line 350 : It can be found that the consistency of the results of the subjects is very poor, and the results are affected by deviations from the mean. The proportion of patients with poor initial values ​​improved more, while the proportion of patients with good initial values ​​improved less. We cannot rule out that this may be just the result of "regression to mean". In fact, with such results, it is more difficult for us to explain that the "intervention" itself really has an impact on gait.

Round 2

Reviewer 1 Report

Comments and Suggestions for Authors

Dear authors, thank you to following my suggestions, the manuscript is better.

As a final remark, all abbreviations in the tables should be explained in the table footnotes, some of them are, but not all of them.

Thank you

Author Response

As a final remark, all abbreviations in the tables should be explained in the table footnotes, some of them are, but not all of them.

The authors appreciate the response and thank the reviewer for their time and contributions to strengthening this manuscript.

As requested, Tables 2, Supplementary Table 1) has been updated to define all abbreviations in the footnote.

Reviewer 2 Report

Comments and Suggestions for Authors

The authors have satisfactorily addressed all comments and suggestions raised during the initial review. The revisions made the manuscript thorough, thoughtful, and effectively address the concerns previously highlighted. The clarity, structure, and overall quality of the manuscript have significantly improved.

In my view, the revised manuscript is now in a publishable state and meets standards expected for publication in "Brain Sciences".

Author Response

The authors have satisfactorily addressed all comments and suggestions raised during the initial review. The revisions made the manuscript thorough, thoughtful, and effectively address the concerns previously highlighted. The clarity, structure, and overall quality of the manuscript have significantly improved.

In my view, the revised manuscript is now in a publishable state and meets standards expected for publication in "Brain Sciences".

Response: The authors appreciate the response and thank the reviewer for their time and contributions to strengthening this manuscript. 

Reviewer 3 Report

Comments and Suggestions for Authors

The authors sought to explore whether extended ante-posterior treadmill training (FLTT/BLTT) could improve walking ability in patients with severe stroke. This may be an important topic, but I still have serious concerns about the study design and the conclusions drawn. According to the research design, the conclusion of this study should be that there is no difference between FLTT and BLTT after 12 training sessions.

Major issue:

  1. Insufficient evidence for cross-study comparisons The main conclusion of this study is that prolonged BLTT (12 sessions) may lead to better outcomes compared with the previously reported 6-session regimen, which relies on an implicit comparison of two different cohorts from different studies. However, the paper did not present any formal comparative analysis, nor did it provide sufficient evidence to prove that the baseline characteristics of the two ethnic groups were comparable. To make this inference valid, the authors should include participants from both studies in a unified analysis or at least provide detailed side-by-side information (e.g., demographics, baseline walking speed, chronic diseases) in a revised Table 1 to justify the comparison.
  2. Although Figure 2 suggests that a subset of participants may have continued to show improvement over the 12 sessions, this alone does not confirm that extending training beyond 6 sessions was the factor causing this phenomenon. The study lacked a control group for 6 sessions, so alternative explanations for the observed benefits—such as natural recovery, a placebo effect, or delayed spontaneous improvement—cannot be ruled out. 
  3. Insufficient mechanistic insights or statistical power The manuscript mentions a "dose-response" assessment, but the small sample size and lack of a control group limit any meaningful dose-response inferences. Furthermore, no clear mechanistic analysis or predictive model was included to determine which individuals might benefit most from extended training.
